# Evaluation of the Relationship between the Weight and Height Percentiles and the Sequence and Chronology of Eruption in Permanent Dentition

**DOI:** 10.3390/healthcare10081363

**Published:** 2022-07-22

**Authors:** Montserrat Diéguez-Pérez, Marta M. Paz-Cortés, Laura Muñoz-Cano

**Affiliations:** 1Preclinical Dentistry Department, Faculty of Biomedical and Health Sciences, European University of Madrid, C. Tajo, s/n, Villaviciosa de Odón, 28670 Madrid, Spain; 2Faculty of Dentistry, Alfonso X El Sabio University, Villanueva de la Cañada, 28691 Madrid, Spain; mmpazcortes@gmail.com; 3Faculty of Dentistry, European University of Madrid, C. Tajo, s/n, Villaviciosa de Odón, 28670 Madrid, Spain; 21110083@live.uem.es

**Keywords:** eruption, dentition, permanent, chronology, sequence, percentile, weight, height

## Abstract

The standard eruption of the permanent dentition in growing patients is influenced by multiple environmental factors. The objective of this research was to study the relationship between height and weight percentiles and the eruption of the permanent dentition. The design of the study was transversal based on the review of the clinical history, visual dental inspection, weight and height indicators, and their respective percentiles in patients in the mixed and definitive dentition stage. The descriptive and comparative analysis of the data was carried out with the statistical software R version 4.1.1. The sample size was 725 participants. The mean age of eruption of the first tooth was 8.0. The eruption sequence in the upper arch was first molar, central and lateral incisor, first premolar, canine, second premolar, and second molar. In the lower arch, the eruption sequence was: central incisor, first molar, lateral incisor, canine, first and second premolar, and second molar. The most frequent weight percentile was P50-97 (50.34%) and height P3-50 (53.38%). Weight (0.0129; 0.0426; 0.0495; 0.000166) and height (0.00768; 0.00473; 0.00927; 10^−5^) variables significantly influenced dental eruption. The factor that most influences the eruption of the permanent dentition is the height percentile.

## 1. Introduction

Dental eruption as well as weight and height percentiles are predictors of general wellbeing in physically growing populations [1,2,3,4,5]. These health predictors are clinically non-invasive, widely used in pediatric practice and methodologically simple and inexpensive indicators useful for large-scale studies [6,7].

The tooth eruption sequence and shedding time are subjects of recent interest worldwide, since they are varying [2,8,9,10,11,12]. The scientific literature considers nutritional status [13], dietary changes, and modifications in masticatory function as causal factors that interact among themselves and may interfere in the short and long term with food absorption and the correct growth and development of the child [12,14,15,16].

The scientific literature provides insufficient information on the relationship between weight and height percentiles and the characteristics of the eruption in permanent dentition [1,3]. Nowadays, clinically, a change in the chronology and sequence of dental eruption has been observed and the latest data available in the Spanish population was published in 2013 [17]. The permanent dentition eruption sequence is an important factor in planning pediatric dentistry and orthodontic treatments [2,5,18]. The exact knowledge of the chronology allows legal and forensic dentistry to determine the chronological age of the child [19]. The percentiles are recorded from birth; therefore, they would early alert on possible eruptive alterations in the future, being able to improve the health and quality of dental care of the child [4,5,6]. The aim was to determine the relationship between height and weight percentiles and the chronology and sequence of eruption of the permanent dentition.

## 2. Materials and Methods

### 2.1. Permits and Approvals

The study was approved by the Regional Ethics Committee of the Community of Madrid on Research with Medicines for clinical research with medical devices regulated by the Royal Decree 1591/2009. It was endorsed by the Research Committee of the European University of Madrid.

### 2.2. Study Design and Regional Area

To achieve the proposed objectives, an epidemiological, observational, descriptive, and cross-sectional study was designed in accordance with the ethical codes established by the World Health Organization.

Data was collected in three pediatric dental care centers in Madrid between November 2020 and September 2021.

### 2.3. Calibration of Examiners

The examiners were thoroughly trained to be able to clinically evaluate tooth eruption and the measurement of weight and height. In a first phase, they received theoretical and preclinical training in order to be qualified to record both parameters. Later, they participated in clinical examinations and discussion of the calibration results, always supervised by a calibrator. Once calibrated, inter- and intra-examiner agreement was calculated for each of the permanent teeth, and the Kappa index obtained was always greater than 0.95.

### 2.4. Sample Size and Selection Criteria

For the calculation of statistical power, the eruption estimation age of the different permanent teeth was taken into account with a precision of 0.25 years and a confidence interval level of 95%, requiring a minimum number of participants in each age group. The sample power was calculated from a minimum of 16 patients per 0.5 years [18]. Thus, after taking into account sex and age range, the minimum total number of patients was 620.

A random sample was selected by establishing the following criteria: 4- to14-year-old Spanish Caucasian boys and girls were included. However, certain participants who had at the time of the study general health problems, alterations in dental development and the absence of a permanent teeth due to any of the following pathologies, such as, cavities, trauma, and non-infectious structural changes, were excluded.

All parents were given the informed consent form for acceptance and the response rate was 100%. In addition to signing the consent form, the children gave their oral approval. 

Due to COVID-19 pandemic, a strategic plan of action was implemented based on the need to fill out a triage questionnaire as a specific measure recommended by the General Council of Spanish Dentists. 

### 2.5. Clinical-Physical Records

According to the established protocols, the patient’s medical history, visual intraoral dental inspection, and growth indicators (weight and height) were recorded, taking into account age and sex. 

All the data collection took place in a dental office with access to a computer and a dental health program, a dental chair with LED light, and a KERN^®^ digital scale that follows the EU’s 90/384/EEC directive. This scale complies with the current European Community regulations for medical use and was previously tested for the determination of mass for medical purposes during control, diagnosis, and treatment.

Due to the Spain’s declaration of the state of alarm during the COVID-19 pandemic, preventive public health measures for dental care in dental practices were implemented. 

None of the patients were uncooperative during the intraoral dental examinations. 

### 2.6. Primary and Secondary Covariates

The chronology and sequence of eruption are the main variables of our study. A maximum of 10 patients per day were examined. At the beginning and in order to identify the permanent teeth erupted at that moment, the systematic approach that was followed was to explain the study objective to the patients and caregivers. Two investigators participated during this clinical phase. 

Percentiles were calculated following a standardized procedure based on weight in kilograms, height in centimeters, and by sex. The patients were asked to remove heavy clothing such as coats and jackets, shoes, and pocket contents. Height was measured with a stadiometer that has an accuracy of 0.01 m. Patients should keep the head aligned with the horizontal plane of Frankfurt. Weight was measured to the nearest 0.1 kg. The patient was carefully positioned in the center of the weighing plate and the mass value was read after having checked a stable value indication. The percentile tables from the United States Centers for Disease Control and Prevention (CDC) [20] were used as reference. Percentile curves were plotted for both normative percentiles: 3rd percentile (P3), 50th percentile (P50), and 97th percentile (P97), and non-normative percentiles such as <3rd and >97th percentiles (P < 3rd and P > 97th). The values obtained were recorded on specific growth charts for weight and height [21] recommended by the WHO (World Health Organization).

### 2.7. Data Analysis

All the information obtained was transferred to a Microsoft Excel^®^ 2021 (Redmond, WA, USA) table where age in months, sex, the permanent teeth present, weight, height, and percentiles were recorded. The analysis was carried out with the R Project for Statistical Computing^®^ (version 4.1.1, Vienna, Austria) using the RStudio integrated development environment. 

During the first stage, a descriptive data analysi s expressed in frequency and contingency tables was performed, with absolute frequencies and percentages. Confidence intervals were used for a proportion. The chi-square test and Fisher’s exact test were used to study the relationship between factors in the contingency tables. Comparison of averages in the two groups was carried out using the *t*-test and the Wilcoxon test. The Pearson’s correlation coefficient and Spearman’s correlation tests were used to analyze the relationship between numerical variables. Results with *p*-values of 0.05 or less were considered significant.

## 3. Results

### 3.1. Characteristics of the Study Population

Eight patients were excluded from the study because they had numeric dental developmental disorders and a personal medical history of Down syndrome and autism spectrum disorder. A pediatric sample of 725 patients distributed by gender: 367 (50.62%) girls and 358 (49.38%) boys were finally analyzed.

### 3.2. Characteristics of Chronology and Eruption Sequence

Table 1 shows the results regarding the chronology and eruption sequence of all permanent teeth, except for wisdom teeth.

Figure 1 shows the box plot of age as a function of the first erupted tooth in each of the hemiarches.

### 3.3. Characteristics of Eruption Symmetry

Regarding the symmetrical eruption sequence, 300 (41.38%) individuals had symmetrical eruption in all hemiarches, while only 2 had it (0.28%) in the upper arch and 28 (3.86%) in the lower arch. The confidence intervals recorded were 37.78–45.07% for symmetry in both arches, 0.05–1.11% for upper symmetry, 2.63–5.61% for lower symmetry, and 50.77–58.14% for eruption asymmetry. A test of proportions showed that the proportion of individuals with symmetry was significantly lower than 50%, with a *p*-value of 2.06 x 10^−6^. In relation to sex, symmetrical eruption was more frequent in girls (156%, 42.51%) than in boys (144%, 40.22%), with no statistically significant differences, obtaining a *p*-value of 0.5832.

### 3.4. Characteristics of Weight and Height Percentiles

The results obtained in reference to percentiles for weight and height are shown in Table 2.

Figure 2 shows the means regarding the relationship between these eruptions’ symmetric eruption and asymmetric eruption, taking into account the different weight and height percentiles. 

Applying the multivariate logistic regression model, it was determined that for every meter increased in height, the probability of symmetrical eruption increased by a factor of 866.75. Additionally, the probability of symmetry was multiplied by a factor of 0.4 for patients with an undesirable percentile. 

Table 3, Table 4, Table 5 and Table 6 show the results regarding the eruption of each of the permanent teeth and percentiles.

## 4. Discussion

### 4.1. Chronology and Sequence

The eruption age range or timing of all permanent teeth varies from the ages of 5 to 13 according to different research studies [22,23,24,25,26,27]. Recently, the trend has changed. The early eruption chronology has been studied, especially in some racial groups [28] and in developed countries. It is believed that these results are possible due to an improvement in nutrition and overall health during childhood, early puberty, and changes in food texture, as well as sexual and socio-economic factors. In our study, the first tooth emerges at 8 years of age, as in Moslemi et al. results [29]. When comparing these findings to results from previous studies in the Spanish population [2,8,9,17], this eruption age is very late, a fact that may be associated with inadequate nutrition [13]. The last tooth to emerge was at 13 years of age and 7 months, exceeding the range established by other authors.

The upper first permanent molar is the first tooth to appear in children whose eruption chronology is delayed. On the other hand, when the eruption chronology is premature, the first tooth to emerge is the lower central incisor. It can be deduced from this that the prompt or delayed eruption is determined by the dental group and may be due to a biomechanical factor. The lower central temporary incisors, being the smallest teeth in the human body, are not capable of supporting the occlusal loads of mastication and exfoliate early, allowing their successors to emerge. Current research shows that frequent meat intake shows a tendency towards early eruption and vegetables towards late eruption [12].

Sex did not affect the eruption of the first tooth [30] except for the eruption that takes place in the lower right hemiarch.

The mean age of eruption of permanent premolars and molars is similar in all the hemiarches, but there were statistically significant differences in the incisor and canine teeth group, the upper teeth emerge at an older mean age when we compare it to the same teeth in the lower arch. 

Logically, age influences significantly during the eruption of permanent teeth per hemiarch (0.00403; 0.00519; 0.0081; 4.09 × 10^−5^) [15]. 

Asymmetrical eruption is slightly more frequent in the male sex, with 214 boys having asymmetrical eruption compared to 211 girls. The proportion of individuals with symmetric eruption is significantly lower than 50%, with a *p*-value of 2.06 × 10^−6^. It is striking that only 4.14% of the population presented asymmetry in only one of the arches. 

As stated by Makino et al. [31], asymmetry is more frequent than expected, especially if other studies are taken into account [2,8,18,19,24,25,26,32,33,34]. In total, 68.62% of the children had asymmetry in both arches. A clear trend is observed between symmetry [35] of eruption in the upper and lower arches (*p*-value < 2.2 × 10^−16^); therefore, the probability of a child having symmetry in the lower arch while he also has upper symmetry (and vice versa) is high. This suggests that masticatory function could be a potential risk factor for asymmetrical eruption as the preferential unilateral chewing side would affect the rhizolysis process of the deciduous teeth in one hemiarch with respect to the contralateral side.

Age as well as gender did not significantly affect the presence of symmetry [2], obtaining a *p*-value of 0.5832. After applying a logistic regression model, the Odds Ratio indicated that both variables had an increasing relationship with symmetry. As age increases, the probability of presenting a symmetrical eruption increases. The probability of eruptive symmetry multiplies by 1.013 for each month lived.

The most frequently erupting first tooth in the upper arch was the first permanent molar, while in the lower arch it was the central incisor [2,8,9,10,11,17,24,36], which differs from other studies [3,14,25,37]. There was a statistically significant relationship between the eruption of the first tooth in the different hemi-arches. This could be due to a functional factor. The chewing in a large number of patients is altered, which explains why there is an increase in the reception of chewing loads in the anterior sector. The lower dental group has a smaller coronal and root dimensions compared to the upper teeth; therefore, eruption begins earlier in the lower arch.

There is a statistically significant relationship between the dental eruption of the central incisor in the fourth quadrant and the gender, as tooth 46 erupts more frequently in boys than in girls. In this hemi-arcade, the probability of the first tooth being 46 is 3.359 times higher for boys than for girls. This finding does not coincide with other Spanish population groups [2,8].

Taking into account each hemiarch, the upper canines and upper second premolars emerge simultaneously; however, in the left hemiarch, the central incisor emerges first, followed by the first permanent molar. On the right side they both emerge simultaneously.

In contrast to other studies [3], this eruption pattern favors the correct development of occlusion and function [2,17,25].

### 4.2. Weight/Height

In contrast to other studies [14,15,38], no statistically significant differences could be found when evaluating the variables age (0.9159), weight (0.4918), and height (0.1647) as a function of sex. 

However, when the patient’s percentile was taken into account, a significant relationship between percentile and sex was observed. Percentiles P < 3 and P < 97 were more frequently present in the male patients. 

With respect to height, no statistical significance was observed between sex and percentile. However, there was significance for mean age as a function of percentile.

### 4.3. Association between Eruption and Weight and Height Percentiles

In contrast to other studies [23], weight (0.0129; 0.0426; 0.0495; 0.000166) and height (0.00768; 0.00473; 0.00927; 10^−5^) variables significantly influence dental eruption [3]. 

Increasing weight and/or height increases the probability of symmetrical dental eruption. For every meter increased in height, the probability of symmetry increases by a factor of 866.75. In fact, it increases significantly in those individuals with a height percentile in the normal range (P3–97). However, this same relationship is not significant for weight percentile.

Furthermore, when comparing the eruption of the first tooth with the weight and height percentiles, a statistically significant relationship was observed between the eruption of the first tooth and height percentile in all four hemi-arcades. A high percentage of individuals with a height percentile in the P < 3 and P > 97 ranges was observed when the first tooth had not yet erupted. On the other hand, no statistical significance was observed between the eruption of first tooth and weight percentile in any quadrant. The probability that the first tooth to erupt was either the first permanent molar or the lower central incisor did not depend on the percentile variable. The probability that the first emerging tooth is the first molar in the upper quadrants does not depend on any percentile. Similarly, the probability that the central incisor is the first tooth to erupt in the lower arch does not depend on the percentiles either. 

Weight percentile does not influence the eruption of any teeth in any quadrant. Nevertheless, the height percentile had a significant relationship with the eruption of central incisors, lateral incisors, and upper first molars. In fact, participants with a percentile outside the normal range (P < 3 and P > 97) did not have these teeth erupted, whereas those children inside the P3, P50, and P97 percentiles did. Regarding the lower central and lateral incisors and lower first molars, individuals with a height percentile outside the normal range did not present them, in contrast to those with a normal percentile (P3, P50, and P97). According to some research, height velocity is an extremely sensitive marker of growth [1]. 

Regarding the strengths of this study, there is a lack of research in the scientific literature that associate these physical-clinical parameters in the pediatric population. In recent years, no data have been published on the eruption of the permanent dentition in the Spanish population. This study shows that percentiles affect not only the chronology but also the eruption sequence. Lower weight and height values can be considered as an etiological factor, so far unknown, of asymmetric eruption. We urge pediatricians to be aware of this association in order to prevent occlusal disorders resulting from pathological eruption.

One limitation of this study is that, since it is a cross-sectional design, it is not possible to include chronological and sequential data on the same patient throughout the entire eruptive process; in fact, this would lead to the loss of patients because the range of observation over time is very wide. Another specification is the unequal distribution of the age of the sample; for this reason, the age is indicated in years with decimals. This limitation is due to the wide age range of the sample.

## 5. Conclusions

We conclude the following aspects:

The age range of eruption in the permanent dentition is between 8 to 13 years and 7 months.

The eruption of the first tooth is determined by the temporal chronology of eruption and the height percentile. In addition, asymmetric eruption is much more frequent than expected and is associated with lower weight and height values.

The weight percentile does not influence the eruption of any teeth in any of the quadrants; however, the fact that participants with a P < 3 and P > 97 height percentile, unlike those with P3, P50, and P97 percentiles, do not show the eruption of the first molar and permanent central and lateral incisors, is relevant.

## Figures and Tables

**Figure 1 healthcare-10-01363-f001:**
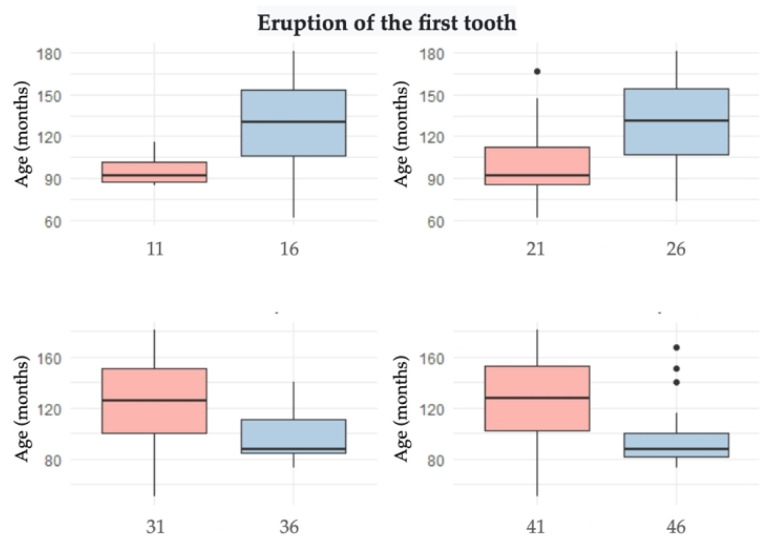
Box plot of age and first tooth.

**Figure 2 healthcare-10-01363-f002:**
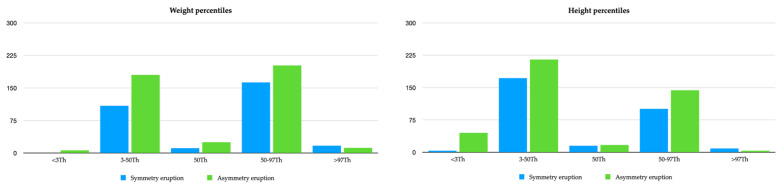
Bar graph of symmetric eruption as a function of percentile.

**Table 1 healthcare-10-01363-t001:** Chronology of the eruption of all the permanent dentition except for the wisdom teeth by age in years and months.

Tooth *	*n*	%	Mean Age (Years and Months)	*p*-Value **
1.1	520	71.7	11.1	4.34 × 10^−8^
2.1	505	69.6	11.1	
3.1	607	83.7	10.4	
4.1	580	80.0	10.6	
1.2	441	60.8	11.6	0.0085
2.2	438	60.4	11.6	
3.2	484	66.7	11.3	
4.2	484	66.7	11.3	
1.3	202	27.8	13.2	1.9 × 10^−11^
2.3	194	26.7	13.2	
3.3	298	41.1	12.5	
43	294	40.5	12.5	
14	268	36.9	12.7	0.0906
24	266	36.6	12.8	
34	236	32.5	13.0	
44	242	33.3	13.0	
15	193	26.6	13.2	0.474
25	191	26.3	13.2	
35	177	24.4	13.4	
45	186	25.6	13.3	
16	556	76.6	10.8	0.836
26	542	74.7	10.9	
36	559	77.1	10.8	
46	558	76.9	10.8	
17	122	16.8	13.6	0.223
27	123	16.9	13.8	
37	148	20.4	13.8	
47	142	19.5	13.6	

* 17 and 27 (right and left upper 2nd molar); 37 and 47 (left and right lower 2nd molar); 16 and 26 (right and left upper 1st molar); 36 and 46 (left and right lower 1st molar); 15 and 25 (right and left upper 2nd premolar); 35 and 45 (left and right lower 2nd premolar); 14 and 24 (right and left upper 1st premolar); 34 and (left and right lower 1st premolar);13 and 23 (right and left upper canine); 33 and 43 (lower right canine); 12 and 22 (right and left upper lateral incisor); 32 and 42 (left and right upper lateral incisor); 11 and 21 (right and left upper central incisor); 31 and 41 (left and right lower central incisor). ** ANOVA test and Welch One-Way test.

**Table 2 healthcare-10-01363-t002:** Contingency table of the different percentiles based on weight and height, sex, and age in years of the sample.

PERCENTILE (Th) (Weight and Height)		<3 Th	3–50 Th	50 Th	50–97 Th	>97 Th	*p*-Value *
n (%) Age (years)	Girls	2 (0.5)	169 (46.0)	15 (4.0)	172 (46.8)	9 (2.4)	0.0031
		7.2	9.4	9.7	10.1	10.5	
	Boys	4 (1.1)	120 (33.5)	21 (5.8)	193 (53.9)	20 (5.5)	
		7.6	9.8	10.0	9.8	9.5	
	Total	6 (0.8)	289 (39.8)	36 (4.9)	365 (50.3)	29 (4.0)	
		7.4	9.6	9.9	10.0	9.8	
	Girls	25 (6.8)	227 (61.8)	13 (3.5)	100 (27.2)	2 (0.5)	2.1 × 10^−5^
		6.6	10.0	10.4	9.8	13.7	
	Boys	24 (6.7)	160 (44.6)	19 (5.3)	145 (40.5)	10 (2.7)	
		6.8	10.1	9.3	9.7	12.2	
	Total	49 (6.7)	387 (53.3)	32 (4.4)	245 (33.7)	12 (1.66)	
		6.7	10.1	9.8	9.8	12.5	

* Fisher’s exact test and *p*-value.

**Table 3 healthcare-10-01363-t003:** Eruption of the incisor group and its relationship with weight and height percentile. Descriptive and comparative statistics: absolute and relative frequencies; contingency tables and Fisher’s exact test.

	Weight Percentiles					Height Percentiles				
	3, 50, and 97		<3 and >97		*p*-Value	3, 50, and 97		<3 and >97		*p*-Value
	No	Yes	No	Yes		No	Yes	No	Yes	
11	192 (27.8)	498 (72.2)	13 (37.1)	22 (62.9)	0.2498	165 (24.8)	499 (75.2)	40 (65.6)	21 (34.4)	2.35 × 10^−10^
21	207 (30.0)	483 (70.0)	13 (37.1)	22 (62.9)	0.3537	181 (27.2)	483 (72.8)	39 (63.9)	22 (36.1)	1.7 × 10^−8^
31	110 (15.9)	580 (84.1)	8 (22.8)	27 (77.2)	0.3443	91 (13.7)	573 (86.3)	27 (44.2)	34 (55.8)	4.97 × 10^−8^
41	137 (19.8)	553 (80.2)	8 (22.8)	27 (77.2)	0.6658	114 (17.7)	550 (82.3)	31 (50.9)	30 (49.1)	1.5 × 10^−8^
12	270 (39.1)	20 (60.8)	14 (40.0)	21 (60.0)	1	244 (36.7)	420 (63.2)	40 (65.6)	21 (34.4)	1.55 × 10^−5^
22	273 (39.5)	417 (60.5)	14 (40.0)	21 (60.0)	1	247 (37.2)	417 (62.8)	40 (65.6)	21 (34.4)	2.8 × 10^−5^
32	227 (32.9)	463 (67.1)	14 (40.0)	21 (60.0)	0.432	201 (30.2)	463 (69.7)	40 (65.6)	21 (34.4)	1.03 × 10^−7^
42	227 (32.9)	463 (67.1)	14 (40.0)	21 (60.0)	0.432	200 (30.1)	464 (69.8)	41 (67.3)	20 (32.7)	1.8 × 10^−8^

**Table 4 healthcare-10-01363-t004:** Eruption of the canines and its relation by hemiarch with weight and height percentiles.

	Weight Percentiles					Height Percentiles				
	3, 50, and 97		<3 and >97		*p*-Value	3, 50, and 97		<3 and >97		*p*-Value
	No	Yes	No	Yes		No	Yes	No	Yes	
13	501 (72.7)	189 (27.3)	22 (62.9)	13 (37.1)	0.2454	476 (71.7)	188 (28.3)	47 (77.1)	14 (22.9)	0.4559
23	508 (73.7)	182 (26.3)	23 (65.8)	12 (34.2)	0.3284	484 (72.9)	180 (27.1)	47 (77.1)	14 (22.9)	0.5476
33	407 (59.0)	293 (41.0)	20 (57.2)	15 (43.8)	0.8613	382 (59.6)	282 (42.4)	45 (37.8)	16 (26.2)	0.0142
43	411 (59.6)	279 (40.4)	20 (57.2)	15 (43.8)	0.8603	386 (58.2)	278 (41.8)	45 (37.8)	16 (26.2)	0.0200

**Table 5 healthcare-10-01363-t005:** Eruption of the premolar group and its relation by hemiarch with weight and height percentiles.

	Weight Percentiles					Height Percentiles				
	3, 50, and 97		<3 and >97		*p*-Value	3, 50, and 97		<3 and >97		*p*-Value
	No	Yes	No	Yes		No	Yes	No	Yes	
14	437 (63.4)	253 (36.6)	20 (57.2)	15 (42.8)	0.4763	412 (62.1)	252 (37.9)	45 (73.8)	16 (26.2)	0.0728
24	439 (63.8)	251 (36.3)	20 (57.2)	15 (42.8)	0.4740	413 (62.2)	251 (37.8)	46 (75.5)	15 (24.5)	0.0511
34	468 (67.9)	222 (37.1)	21 (60.0)	14 (40.0)	0.3574	444 (66.9)	220 (31.1)	45 (73.8)	16 (26.2)	0.3185
44	462 (66.8)	228 (33.2)	21 (60.0)	14 (40.0)	0.4625	437 (65.9)	227 (34.1)	46 (75.5)	15 (24.5)	0.1558
15	507 (73.5)	183 (25.5)	21 (71.5)	10 (28.5)	0.8447	484 (72.9)	180 (27.1)	48 (78.7)	13 (21.3)	0.3669
25	510 (74.6)	180 (25.4)	24 (68.6)	11 (31.4)	0.5549	488 (73.5)	176 (26.5)	46 (75.6)	15 (24.4)	0.8794
35	526 (76.3)	164 (23.7)	22 (62.9)	13 (37.1)	0.1038	502 (75.6)	162 (24.4)	46 (75.6)	15 (24.4)	1
45	516 (74.8)	174 (52.2)	23 (65.8)	12 (34.2)	0.2368	493 (74.3)	171 (25.7)	46 (75.6)	15 (24.4)	1

**Table 6 healthcare-10-01363-t006:** Eruption of the molar group and its relation by hemiarch with weight and height percentiles.

	Weight Percentiles					Height Percentiles				
	3, 50, and 97		<3 and >97		*p*-Value	3, 50, and 97		<3 and >97		*p*-Value
	No	Yes	No	Yes		No	Yes	No	Yes	
16	159 (21.0)	531(77.0)	10 (28.5)	25 (71.5)	0.4197	134 (23.1)	531 (76.9)	35 (58.4)	25 (41.6)	1.84 × 10^−9^
26	172 (24.9)	518 (75.1)	11 (31.4)	24 (68.6)	0.4250	114 (21.6)	520 (78.4)	39 (63.9)	22 (36.1)	2.3 × 10^−11^
36	156 (22.6)	534 (77.4)	10 (28.5)	25 (71.5)	0.4122	130 (19.5)	534 (80.5)	36 (59.1)	25 (40.9)	1.8 × 10^−10^
46	157 (22.7)	553 (77.3)	10 (28.5)	25 (71.5)	0.4145	129 (19.4)	635 (80.6)	38 (62.3)	23 (37.7)	5.5 × 10^−12^
17	574 (83.2)	116 (16.8)	29 (82.9)	6 (17.1)	1	552 (83.2)	112 (16.8)	51(83.7)	10 (16.3)	1
27	573 (83.1)	117 (16.9)	29 (82.9)	6 (17.1)	1	554 (83.5)	110 (16.5)	48 (78.7)	13 (21.3)	0.3719
37	553 (80.2)	137 (19.8)	24 (68.6)	11 (31.4)	0.129	528 (79.6)	136 (20.4)	49 (80.4)	12 (19.6)	1
47	558 (80.9)	132 (19.1)	25 (71.5)	10 (28.5)	0.189	534 (80.5)	130 (19.5)	49 (80.4)	12 (19.6)	1

## Data Availability

The data presented in this study are available on request from the corresponding author (M.D.-P.).

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
