# Peer review of "Evaluation of the Relationship between the Weight and Height Percentiles and the Sequence and Chronology of Eruption in Permanent Dentition"

_healthcare, 2022, doi:10.3390/healthcare10081363_

Round 1

Reviewer 1 Report

Line 33 - I believe 13 is a reference but it is not formatted as such.

Line 39 - Please add a reference of the published data on the Spanish population

Lines 36-45 should have some references.

Line 45 - "The main aim"? What were the other aims?

The WHO describes sex as characteristics that are biologically defined, whereas gender is based on socially constructed features. As so, I believe that children sex was recorded; not the gender.

Lines 97-98 - "A maximum of 10 patients were examined" - ?? per day...?

Line 108 - please add a reference for the CDC tables

Line 116 - excel should be correctly cited with its full name. Brand and country should be provided. 

Line 118 - Please add brand and country for each of the software used.

Table 2 - please add an explanation for Rd and Th.

What are the implications of this study? Are there any clinical considerations? Any recommendations for the reader?

Author Response

Dear Reviewer:
The authors are grateful for your comments and suggestions, which have helped to improve the way we present our research. We will detail and resolve each of your comments below.
We kindly ask you to accept your responses.

Reviewer 2 Report

The study used different analyses to determine the relationship between height and weight percentiles and the chronological order of eruption of the permanent dentition. The objective is clear, and the study is meaningful for the clinics.

However, I have some questions.

1. Table 1. Chronology of the eruption of all the permanent dentition except for the wisdom teeth by age in months. Then in the table, it’s shown as Mean Age (years and months).

Table 2. Contingency table of the different percentiles based on weight and height, sex and age in months of the sample. But in the table, it’s shown as Age (years). It would be better to be consistent.

2. In table 3, 4 and 5, Height percentiles <3 and >97 was analyzed together as a group. As the study showed that “The factor that most influences the eruption of the permanent dentition is the height percentile”, and <3 and >97 can represent two groups, one (<3) is a bit delayed and the other (>97) is an overgrown state, so I suggest that it is better to separate them into different groups for analysis.

For similar reasons, I also suggest that it is better to separate <3 and >97 into different groups for analysis of Weight percentiles.

3. There are some errors or typos, Line 33, “13” may be the reference number.

Author Response

Dear Reviewer:
The authors are grateful for your comments and suggestions, which have helped to improve the way we present our research. We will detail and resolve each of your comments below.
We kindly ask you to accept your responses.

This manuscript is a resubmission of an earlier submission. The following is a list of the peer review reports and author responses from that submission.